# Plant Functional Diversity Is Linked to Carbon Storage in Deciduous Dipterocarp Forest Edges in Northern Thailand

**Lamthai Asanok** [1],*, **Rungrawee Taweesuk** [2] **and Torlarp Kamyo** [1]

1   Department of Agroforestry, Phrae Campus, Maejo University, Phrae 54140, Thailand; torlarp66@gmail.com
2   National Science and Technology Development Agency, Bangkok 10400, Thailand;
    rungrawee.taw@nstda.or.th
*   Correspondence: lamthainii@gmail.com; Tel.: +66-89-014-5461

**Abstract:** Studies of carbon storage using functional traits have shown that it is strongly affected by functional diversity. We explored the effects of functional diversity on carbon storage at the edge of a deciduous dipterocarp forest (DDF) ecosystem in Thailand. Aboveground biomass carbon (AGBC), soil organic carbon (SOC), and total ecosystem carbon (TEC) were used as indicators of carbon storage. Five functional traits were measured in 49 plant species to calculate the community-weighted mean (CWM) and Rao's quadratic diversity (FQ). We assessed which functional diversity metrics best-explained carbon storage. The results indicated that AGBC had a significant, positive relationship with the FQ of wood density, and a negative relationship with the CWM of leaf thickness. SOC had a significant, negative association with the FQ of leaf thickness and a positive relationship with the CWM of specific leaf area (SLA). TEC was best predicted by increases in the FQ of wood density and the CWM of SLA. These findings indicate that CWM and FQ are important for understanding how plant traits influence carbon storage in DDF edge ecosystems and suggest that promoting a high diversity of species with dissimilar wood density and high SLA may increase carbon storage in chronically disturbed DDF ecosystems.

**Keywords:** seasonally dry tropical forest; ecosystem functioning; community-weighted mean; Rao's quadratic diversity; species diversity



## 1. Introduction

The increasing atmospheric concentration of carbon dioxide is important to global climate change [1]. Forests are an excellent means of capturing atmospheric carbon via carbon sequestration [2]. Carbon storage in a forest at any given time reflects the net balance between carbon uptake, loss, and storage processes [3], and the ability of plants to capture carbon from the atmosphere was assessed by estimating carbon storage in forests [4]. Recently, the carbon storage capacity of forests was empirically estimated using trait-based approaches [5,6]. Functional diversity is significantly associated with ecosystem processes and can be used to estimate carbon storage in forest ecosystems [5]. The relationship between trait values and plant diversity was used to explain the linkages between plant functional diversity and carbon accumulation in forests [6,7]. These relationships are attributable to trade-offs between resource acquisition and conservation [8], because acquisitive traits may promote higher carbon flows, whereas conservative traits may be conducive to higher carbon storage at the ecosystem level [9–11]. These trade-offs have resulted in two contrasting hypotheses, the mass ratio and niche complementarity hypotheses, that predict the relationship between functional traits and carbon stocks in natural forest ecosystems [12,13]. The mass ratio hypothesis assumes that the trait values of individual species affect their relative abundance and that the dominance of plant functional traits at the community level can be estimated using the community-weighted mean [12]. The niche complementarity hypothesis posits that complementarity effects, through variations in resource-use strategies, promote efficient use of resources

by functionally diverse species [14], and can be estimated using functional diversity indices [5]. These two hypotheses are differently significant under different circumstances in natural ecosystems.

Deforestation in tropical regions is a major source of greenhouse gases [15] and predicting ecosystem carbon storage is important for estimating the impacts of forest loss [16]. Deciduous dipterocarp forests (DDFs), defined as seasonally dry tropical forests dominated by deciduous dipterocarp species, occur in tropical monsoon climates where the dry season lasts at least 4–5 months, and are widespread in South and Southeast Asia [17,18]. This forest type has a long history of chronic disturbance from frequent fires and human activities [19]. Agricultural practices, such as slash-and-burn and highland agriculture, have resulted in the loss, degradation, and fragmentation of DDF ecosystems throughout Thailand [20], increasing the extent of agricultural land, forest edge communities, and remnant forests in headwater areas [21]. As a result, forest edges in the vicinity of headwater areas are in urgent need of protection, which may also enhance the conservation of adjacent forest remnants. Carbon storage is among the ecosystem services used to generate support for forest conservation strategies [22]; evaluating the carbon storage capacity of forest edges, including DDF remnants, may therefore encourage the conservation of these ecosystems. Tree carbon stocks may be higher in forest interiors than at edges and are positively related to interior fragment size [23]. Carbon stocks may be, on average, up to 25% lower on edges than in forest interiors, reflecting carbon loss due to forest fragmentation [24]. The aim of this study was thus to gain an understanding of the relationship between carbon storage in different components of trees in an edge of DDF remnants, using the community-weighted mean (CWM) and functional diversity (i.e., Rao's quadratic diversity index, FQ) of various plant traits. Our objective was to investigate the effects of plant functional diversity on carbon storage at the edges of DDF remnants with respect to responses to forest fragmentation. We expected that by combining data on carbon storage with functional trait data, we would generate new insights into ecosystem function in degraded areas.

## 2. Materials and Methods

### 2.1. Study Area

The study was conducted along the edge of DDF remnants (i.e., the transition between the edge and the forest interior) in the Mae Khum Mee sub-watershed area (18°22′–18°28′N, 100° 08′–100°33′E). This region covers 452.4 km² of Phrae Province in northern Thailand (Figure 1), and the site spans elevations of 320–540 m above mean sea level (a.m.s.l.). Mean annual temperature and rainfall are 26.5 °C and 1400 mm, respectively. The region experiences two main seasons: a wet season (May–October; mean rainfall and temperature of 1460 mm and 27.51 °C) and a dry season is subdivided into cool-dry (November–January; mean rainfall and temperature of 110 mm and 22.56 °C) and hot-dry sub-seasons (February–April; mean rainfall and temperature of 406 mm and 27.16 °C). The numerous large remnant DDF patches in the headwaters area regularly experience wildfires during the dry season as a result of farmers burning corn stumps after harvest. The canopy is dominated by deciduous species such as *Shorea obtusa*, *Shorea siamensis*, *Dipterocarpus obtusifolius*, *Mitragyna rotundifolia*, *Lannea coromandelica*, *Aporosa nigricans*, *Dalbergia oliveri*, *Strychnos nux-vomica*, *Terminalia chebula*, and *Buchanania lanzan* [25].

### 2.2. Sampling Plot Selection and Tree Data

From January to December 2017, plots along the edge-interior gradients were established within the transition zone from the edge into the forest interior of the DDFs area. We selected sites with similar topographic attributes, e.g., altitudes of 400 m a.m.s.l. and a slope of 45%. Maize had been cultivated at all sites prior to abandonment 3 years before sampling, as confirmed by interviews with local residents. We established three 10 m × 100 m belt plots at the site. Each belt plot was oriented perpendicular to the forest edge and spanned 100 m from the edge to the interior (Figure 2); the first plot was located near the first mature tree at the edge. Each belt plot was subdivided into ten 10 m × 10 m

plots, for a total of 30 plots. In each 10 m × 10 m plot we measured the height and diameter at breast height (DBH) of all trees ≥1.3 m in height and ≥4.5 cm in DBH. Tree height was measured using a range finder (Nikon Forestry Pro II), and DBH was measured using a diameter tape. All trees were identified to the species level by collecting specimens and comparing them to standard specimens at the herbarium of the National Park, Wildlife and Plant Conservation, Thailand. Nomenclature follows the Flora of Thailand [26].

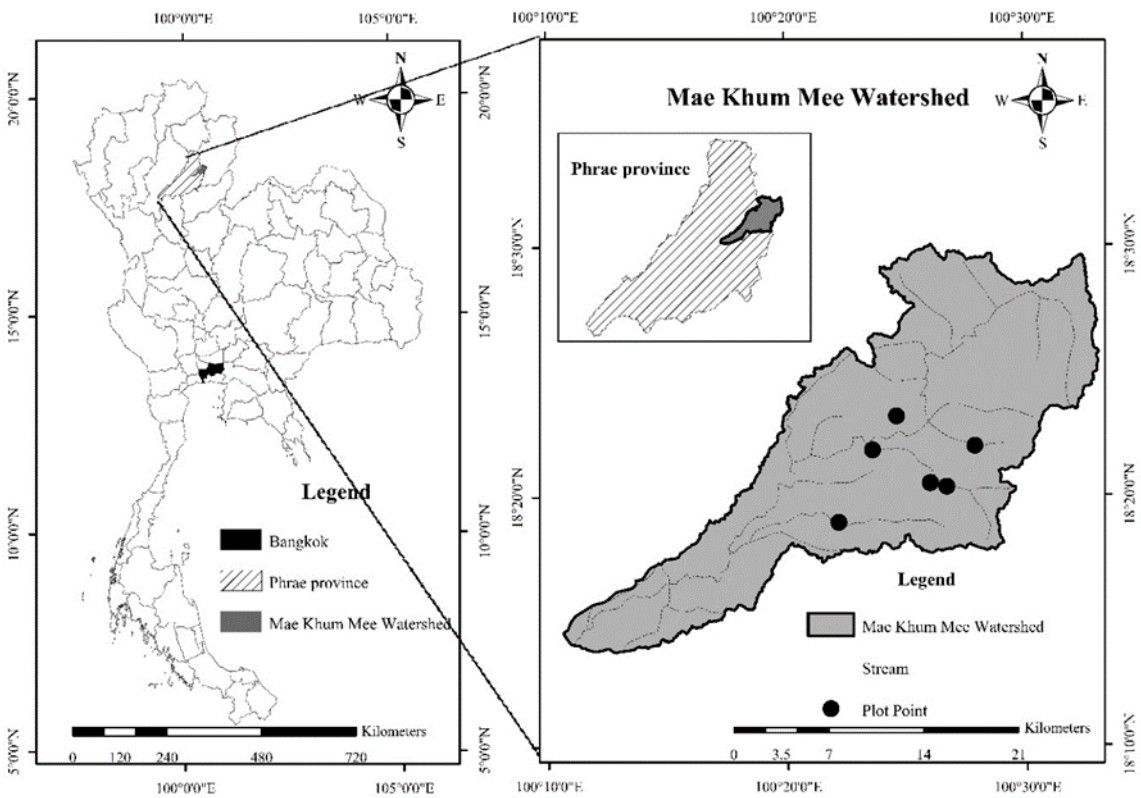

**Figure 1.** The Mae Khum Mee sub-watershed and study sites in Phrae Province, northern Thailand.

### 2.3. Carbon Storage Assessment

We used three indicators of carbon storage: aboveground biomass carbon of trees (AGBC), soil organic carbon (SOC), and total ecosystem carbon (TEC, i.e., AGBC + SOC). We excluded the biomass of leaf litter and understory plants from our analyses because the study site regularly experiences wildfires during the dry season.

Aboveground biomass (AGB) was quantified based on the trees measured in all 30 plots and was used to quantify AGBC. We estimated biomass using the allometric model for DDF [27], and estimated the AGB (Mg ha$^{-1}$) of individual trees as follows:

$$WS = 0.0396(D^2H)^{0.9326} \tag{1}$$

$$Wb = 0.006003(D^2H)^{1.027} \tag{2}$$

$$Wl = (28/(Ws + Wb + 0.025^2))^{-1} \tag{3}$$

$$Wt = WS + Wb + Wl \tag{4}$$

where D is DBH (cm), H is tree height (m), Ws, Wb, and WL are biomass (kg) of stems, branches, and leaves, respectively, and Wt is total ABG (kg).

AGBC (Mg C ha$^{-1}$) was estimated using the Intergovernmental Panel on Climate Change conversion rate for biomass to carbon of 0.47 [28], as follows:

$$AGBC = 0.47 \times AGB \tag{5}$$

SOC was quantified by collecting 100 cm$^3$ soil samples from the topsoil (0–15 cm) using a soil core sampler. We collected subsamples from the center and the four corners of each 10 m × 10 m plot, for a total of five points per plot. Two sets of soil samples were collected per plot. The first set was used to assess soil bulk density (SDb; g cm$^{-3}$), which was quantified based on sample mass after oven drying divided by the total sample volume. Subsamples ($n = 5$) were averaged to obtain a single measurement per plot. Samples from the second set were air-dried, sieved using a 2-mm mesh, and analyzed for organic carbon (OC, g kg$^{-1}$) using standard procedures [29,30]. OC was averaged by plot (n = 5), and SOC (Mg C ha$^{-1}$) was calculated as follows:

$$SOC = OC \times SDb \times soil\ depth \tag{6}$$

Finally, we calculated TEC by adding AGBC and SOC. All carbon storage components were used to analyze carbon–trait relationships.

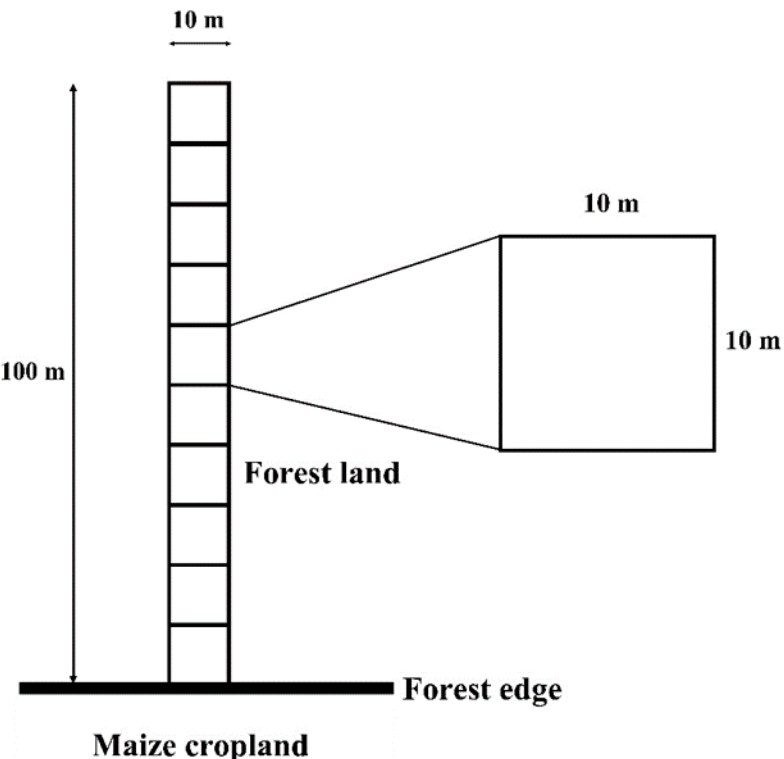

**Figure 2.** Belt plots (10 m × 100 m) were established in a deciduous dipterocarp forest (DDF) in the transition area from the edge to the interior of the forest.

### 2.4. Plant Trait Measurements

We selected functional traits expected to affect carbon storage in forest ecosystems [5,6]. Five stem and leaf traits were included in analyses: specific leaf area (SLA; cm$^2$ g$^{-1}$), leaf dry matter content (LDMC; mg g$^{-1}$), leaf area (LA; cm$^2$), leaf thickness (LT; mm), and wood density (WD; g cm$^{-3}$). Tree species represented by ≥3 individuals with a DBH ≥ 4.5 cm across the 30 plots were selected for trait assessment. We collected sun-leaf samples from each species at the end of the wet season in October 2017 to calculate leaf traits (SLA, LDMC, LA, and LT). We randomly sampled two to ten leaves from three individuals of each species and measured traits following standard methods [31]. Fresh leaves were

scanned using Image*J* software (http://rsbweb.nih.gov/ij/, accessed on 2 September 2021), and images were used to calculate LA. Leaf mass was recorded before and after drying at 60 °C for 48 h to a constant weight. SLA (cm$^2$ g$^{-1}$) was calculated based on the ratio of fresh LA to leaf mass after oven drying. LDMC (mg g$^{-1}$) was calculated based on the ratio of dry mass to the fresh mass of individual leaves. LT (mm) was estimated based on the mean leaf blade thickness of five leaf samples, as measured using a thickness gauge (China YH-1; Zhejiang Top Cloud-Agri Technology, Zhejiang, China). To determine WD, we collected core samples at breast height from the same individuals used for leaf sampling, using a 5-mm increment borer. WD (g cm$^{-3}$) was calculated based on the ratio of mass (following oven drying) to fresh core volume (diameter = 0.5 cm, length = 5 cm).

### 2.5. Functional Diversity Assessment

We estimated functional dominance (i.e., CWM) and divergence (i.e., Rao's quadratic diversity; FQ) for plant traits and compared these metrics to carbon storage in different components (i.e., ABGC, SOC, and TEC). CWM measures the functional identity of dominant species, whereas FQ is a measure of functional dissimilarity among species.

Dominant trait values in each plot were represented by CWM. This metric represents the expected functional trait values of a specific community, and is calculated as follows:

$$\text{CWM} = \sum_{i=1}^{n} p_i \times tr_i \tag{7}$$

where $p_i$ and $tr_i$ are the relative abundance and trait values for species $i$, respectively, and $n$ is the number of species in the plot.

Divergence in individual trait values was represented as functional trait divergence. We quantified trait divergence using Rao's quadratic diversity index (FQ) [32,33]. Rao's quadratic entropy represents community trait divergence based on the sum of dissimilarities among all possible pairs of species in trait space, and is weighted by the product of species' relative abundances, as follows:

$$FQ = \sum_{1-1}^{s-1} \sum_{j=i+1}^{s-1} dijPiPj \tag{8}$$

$$dij = \sum_{t=1}^{T} (Xtj - Xti)2 \tag{9}$$

where *dij* is the Euclidian dissimilarity between the traits of each pair of species *i* and *j*, *Xti* is the trait value of *i*th species, and *t* is the number of traits.

Both CWM and FQ are single-trait indices and use the relative abundance of species to estimate their contribution to community-level variability. CWM and FQ were calculated using the FD package [34] in R version 3.4.1 (R Development Core Team, 2017).

### 2.6. Statistical Analyses

Descriptive statistics were used to explore each component of tree dimensions (DBH, tree height, and biomass), carbon storage (AGBC, SOC, and TEC), and functional diversity (CWM and FQ). We used the procedure to test the relationship between carbon storage and functional diversity [5]. First, we used simple linear regressions to test for pairwise relationships between carbon storage and functional diversity components and identify functional diversity variables that were significantly associated. All significant correlations were assessed using the residuals and predicted values (Supplementary Material Figure S1). Next, we used stepwise multiple regression to test the relationship between carbon storage and functional diversity indices. We selected variables for inclusion in stepwise multiple regression that had Pearson correlation coefficients of <0.7 and tested for the variance inflation factor using the step AIC function of the MASS package in R version 3.6.2. The best equation for each carbon storage component was selected based on metrics such

as adjusted $R^2$ (Adj. $R^2$) and standard error (SE) of the model [35]. The best-supported model represented the relationship between carbon storage components and the functional diversity index.

## 3. Results

Tree diameter at breast height varied between 4.46 and 36.46 cm and tree height was highly variable (2–14 m). Accordingly, biomass was also highly variable (14.65–3398.73 kg per stem; Table 1). Aboveground biomass carbon was highly variable among plots (90–545 Mg C ha$^{-1}$; Table 1). Soil organic carbon varied between 23 and 206 Mg C ha$^{-1}$, and Total ecosystem carbon varied between 147 and 762 Mg C ha$^{-1}$ (Table 1). The estimated community weighted mean (CWM) represented the dominance of traits between plots (Table 1). The CWM of LA was 99–586 cm$^2$, whereas that of SLA was 83–137 cm$^2$ g$^{-1}$. The CWM of LT was 0.17–0.35 mm, and that LDMC was 405–573 mg g$^{-1}$. The CWM of WD was 0.19–0.85 g cm$^{-3}$ (Table 2). Rao's quadratic diversity index (FQ) of all traits varied among plots (Table 1). The FQ of LA was 0.01–2.05 cm$^2$, and that of SLA was 0.02–1.49 cm$^2$ g$^{-1}$. The FQ of LT was 0.12–1.55 mm, and that of LDMC was 0.01–1.40 mg g$^{-1}$. The FQ of WD was 0.03–1.51 cm$^2$ (Table 1).

**Table 1.** Descriptive statistics for tree dimension variables, components of carbon storage; above-ground biomass carbon (AGBC]), soil organic carbon (SOC), and total ecosystem carbon (TEC) and functional diversity; Rao's quadratic diversity index (FQ) and community weighted mean (CWM) of specific leaf area (SLA), leaf dry matter content (LDMC), leaf area (LA), leaf thickness (LT), and wood density (WD) in the deciduous dipterocarp forest edge ecosystem.

| Component of Trees, Carbon and FD | Minimum | Maximum | Mean ± SD |
|---|---|---|---|
| Tree dimension | | | |
| DBH (cm) | 4.46 | 36.46 | 11.93 ± 5.48 |
| Tree height (m) | 2.00 | 14.00 | 8.29 ± 2.39 |
| Biomass (kg stem$^{-1}$) | 14.65 | 3398.73 | 4.10.69 ± 473.11 |
| Carbon component | | | |
| AGBC (Mg $C$ ha$^{-1}$) | 90.04 | 545.90 | 320.6 ± 20.34 |
| SOC (Mg $C$ ha$^{-1}$) | 23.35 | 206.64 | 48.67 ± 5.83 |
| TEC (Mg $C$ ha$^{-1}$) | 147.33 | 762.84 | 455.84 ± 27.04 |
| FQ | | | |
| FQ-LA | 0.02 | 2.05 | 0.67 ± 0.11 |
| FQ-SLA | 0.03 | 1.49 | 0.42 ± 0.06 |
| FQ-LT | 0.12 | 1.55 | 0.83 ± 0.07 |
| FQ-LDMC | 0.01 | 1.40 | 0.47 ± 0.07 |
| FQ-WD | 0.03 | 1.52 | 0.69 ± 0.08 |
| CWM | | | |
| CWM-LA | 99.07 | 586.08 | 258.61 ± 22.33 |
| CWM-SLA | 83.18 | 137.88 | 105.25 ± 2.45 |
| CWM-LT | 0.18 | 0.35 | 0.29 ± 0.01 |
| CWM-LDMC | 405.71 | 573.46 | 469.21 ± 6.55 |
| CWM-WD | 0.19 | 0.85 | 0.55 ± 0.03 |

**Table 2.** Simple linear regressions between components of carbon storage; aboveground biomass carbon (AGBC), soil organic carbon (SOC), and total ecosystem carbon (TEC) and functional diversity; Rao's quadratic diversity index (FQ) and community weighted mean (CWM) of specific leaf area (SLA), leaf dry matter content (LDMC), leaf area (LA), leaf thickness (LT), and wood density (WD) in a deciduous dipterocarp forest edge ecosystem.

| Functional Diversity Component | AGBC | | | SOC | | | TEC | | |
|---|---|---|---|---|---|---|---|---|---|
| | Estimate | $R^2$ | *p*-Value | Estimate | $R^2$ | *p*-Value | Estimate | $R^2$ | *p*-Value |
| FQ of LA | 2.401 | 0.013 | 0.945 | −7.136 | 0.019 | 0.469 | −4.086 | <0.001 | 0.925 |
| FQ of SLA | 85.556 | 0.260 | 0.166 | 12.223 | 0.017 | 0.495 | 120.879 | 0.076 | 0.140 |
| FQ of LT | 80.431 | 0.077 | 0.139 | −28.9 | 0.121 | 0.061 | 73.247 | 0.036 | 0.316 |
| FQ of LDMC | 10.533 | <0.001 | 0.857 | 14.431 | 0.027 | 0.385 | 27.809 | 0.005 | 0.720 |
| FQ of WD | 139.979 | 0.337 | 0.001 | 25.962 | 0.142 | 0.061 | 203.735 | 0.405 | <0.001 |
| CWM of LA | −0.221 | 0.059 | 0.196 | −0.035 | 0.018 | 0.481 | −0.316 | 0.068 | 0.164 |
| CWM of SLA | 2.671 | 0.104 | 0.083 | 1.224 | 0.266 | 0.004 | 4.616 | 0.175 | 0.021 |
| CWM of LT | −879.284 | 0.119 | 0.061 | −276.472 | 0.144 | 0.039 | −1393.16 | 0.17 | 0.024 |
| CWM of LDMC | 0.060 | 0.019 | 0.919 | −0.027 | 0.001 | 0.875 | 0.050 | <0.001 | 0.965 |
| CWM of WD | −297.92 | 0.202 | 0.013 | −77.604 | 0.167 | 0.025 | −455.963 | 0.268 | 0.003 |

Simple linear regressions between the carbon components and FD indices indicated that carbon components were more strongly associated with CWM than FQ. WD, LT, and SLA had significant, positive and negative associations with carbon storage components (Table 2). AGBC was positively associated with increases in the FQ of WD but negatively associated with the CWM of WD (Figure 3A,B), suggesting increasing variance with divergence in WD. These relationships demonstrate that both the FQ and CWM of WD significantly impact biomass carbon. SOC was only significantly associated with changes in CWMs (Figure 3C–E). SOC was positively associated with the CWM of SLA but negatively associated with the CWM of LT and WD (Figure 3C,D), indicating that the dominance of traits such as SLA, LT, and WD may positively or negatively influence soil carbon. TEC also had a significant, positive association with the CWM of SLA and the FQ of WD (Figure 3F,I). By contrast, TEC was negatively associated with the CWM of LT and WD (Figure 3G,H), indicating that the functional dominance and divergence of leaf and wood traits, respectively, contribute significantly to carbon storage in DDF edge ecosystems. We then selected functional diversity variables that best-predicted carbon storage components; these included the FQ of WD and the CWMs of SLA, LT, and WD.

All functional diversity components that significantly predicted carbon storage were combined in stepwise multiple regression models (Table 3). AGBC had a significant, positive association with the FQ of WD and a significant, negative association with the CWM of LT, suggesting that standing biomass carbon was affected by the dissimilarly of WD among species and the low abundance of thick-leaved species in the community. SOC had a significant, positive association with the CWM of SLA and a significant, negative association with the FQ of LT, suggesting that SOC accumulation was impacted by the abundance of species with high SLA and the low diversity of thick-leaved species. TEC was positively associated with the FQ of WD and the CWM of SLA (Table 3), suggesting that TEC stocks were influenced by both variations in WD among species, and an abundance of species with high SLA.

**Table 3.** Final models obtained from stepwise multiple linear regressions between components of carbon storage; aboveground biomass carbon (AGBC), soil organic carbon (SOC), and total ecosystem carbon (TEC) and functional diversity; Rao's quadratic diversity index (FQ) and community weighted mean (CWM) of specific leaf area (SLA), leaf thickness (LT), and wood density (WD) in a deciduous dipterocarp forest edge ecosystem.

| Response Variable | Model Form | Predictor Variables | Slope | SE | *p*-Value | Adj. R² |
|---|---|---|---|---|---|---|
| AGBC | AGBC = 224.10 + 139.98 × FQ_WD − 28.90 × CWM_LT | Model | | 92.3 | 0.0007 | 0.313 |
| | | FQ-WD | + | | 0.0081 | |
| | | CWM-LT | − | | 0.0038 | |
| SOC | SOC = 80.12 − 45.11 × FQ_LT + 3.22 × CWM_SLA | Model | | 27.83 | 0.0035 | 0.542 |
| | | FQ-LT | − | | 0.0012 | |
| | | CWM-SLA | + | | 0.0002 | |
| TEC | TEC = 110.03 + 232.57 × FQ_WD + 4.62 × CWM_SLA | Model | | 90.40 | 0.0008 | 0.403 |
| | | FQ-WD | + | | 0.0059 | |
| | | CWM-SLA | + | | 0.0035 | |

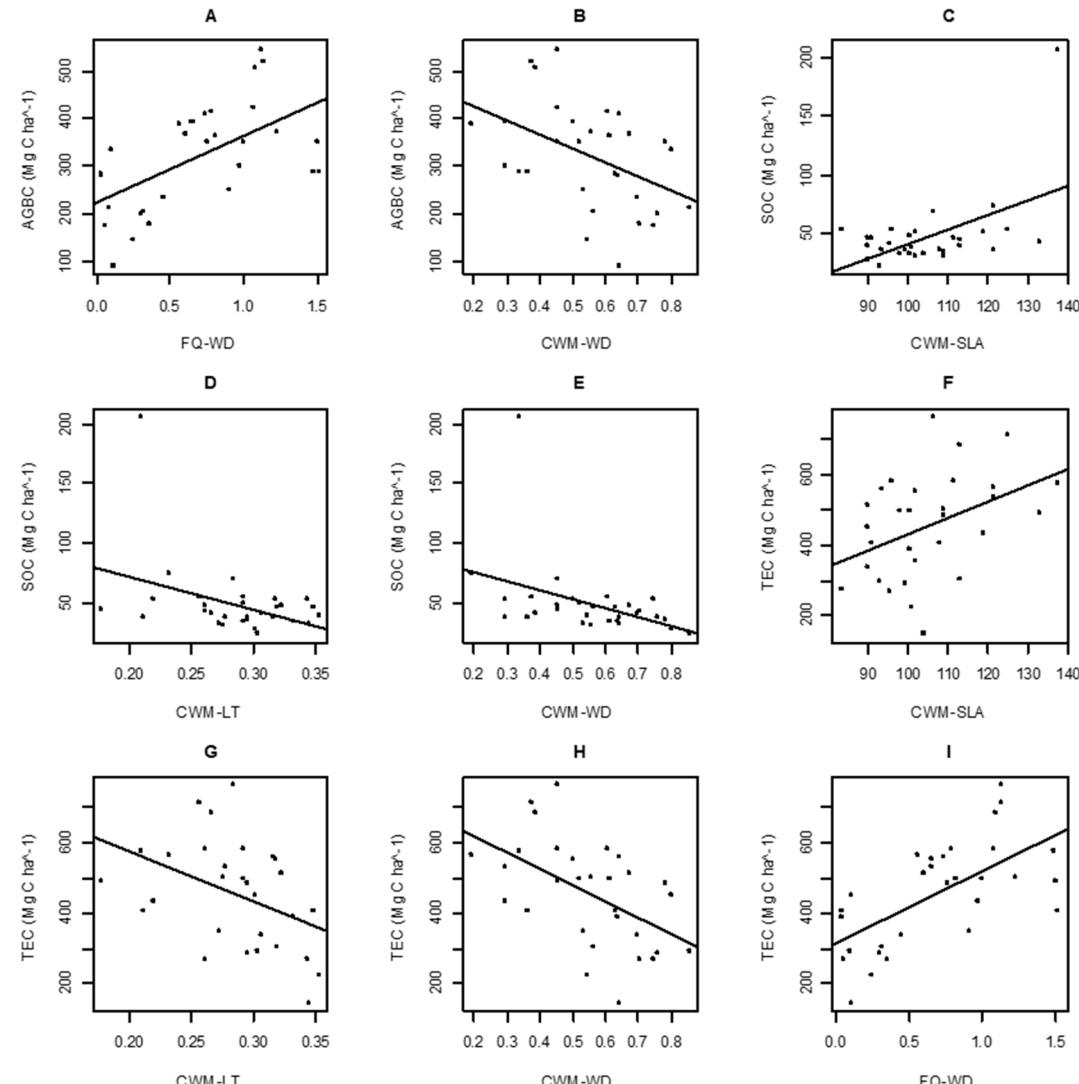

**Figure 3.** Bivariate relationships between components of carbon storage; aboveground biomass carbon (AGBC), soil organic carbon (SOC), and total ecosystem carbon (TEC) and functional diversity; Rao's quadratic diversity index (FQ) and community weighted mean (CWM) of specific leaf area (SLA), leaf thickness (LT), and wood density (WD). Only variables included in the multiple linear regression analysis are shown (see Table 2).

## 4. Discussion

Relationships between carbon storage and functional traits can be used to infer forest ecosystem processes under a changing climate. The functional diversity of plant communities (i.e., the divergence and dominance of functional traits) can predict the influence of ecosystem functions such as carbon storage [5]. We used Rao's quadratic diversity (FQ) as an index of functional divergence because it captures aspects of community structure, such as the mean and dispersion of functional traits, within a given species assemblage [32]. High divergence is a consequence of high values of functional traits among the most abundant species in the community [36], suggesting that strong competition between species leads to niche variation, high trait complementarity, and resource-use differentiation. These processes affect biomass production in forest ecosystems [13,36]. CWM values represent functional dominance, as they reflect the traits of dominant species within an assemblage [12]. The effects of forest edge habitat on functional diversity were not linear, but depended on trait type, indicating that different trait diversity components may respond differently to disturbance [37,38]. Previous studies have suggested that anthropogenic edges affect functional diversity in the highly diverse tree communities of tropical rainforest and temperate mixed forests [38,39] and that enhanced carbon stocks may be found in forest edge habitats [23,24]. Here, the best-supported models indicated that both CWM and FQ contributed to explaining the three carbon storage components in the DDF edge ecosystem. Previous studies have indicated that this finding supports both the mass ratio and niche complementarity hypotheses [5,6,40].

AGBC storage increased with higher divergence in WD, and lower abundance of thick-leaved species. This was because the WD value distribution in plots with high variation species was concentrated toward high values of AGBC. In other words, plots with less homogenous LT values were dominated by thicker-leaved species. As a result, AGBC was influenced by divergence in stem traits, but dominance in leaf traits. This finding indicates that conservative trait syndromes (i.e., high WD) are conducive to high carbon storage capacity in aboveground ecosystem components. This is because DDF forests are dominated by deciduous dipterocarp species such as *Shorea obtusa*, *Shorea siamensis*, and *Dipterocarpus obtusifolius*, which produce dense wood [41]. The diversity of such species promoted dissimilarity in WD. High WD indicates higher efficiency in constructing tissues per wood volume, thus increasing aboveground biomass [42], which is strongly correlated with carbon storage [43]. By contrast, the model indicated a negative relationship with the CWM of LT, suggesting that the number of thick-leaved species at the site was low. Because DDF edges tend to be chronically disturbed, species diversity was high, but the density of each individual species was low [25].

SOC storage may be promoted by the lower divergence of conservative leaf traits, such as LT, and dominance of acquisitive leaf traits such as SLA. Other studies have highlighted the influence of CWM and functional divergence on SOC storage [6,44,45], suggesting that leaf traits affect soil carbon accumulation. This effect is attributable to changes in nutrient availability resulting from the breakdown of leaves, which in turn affects the OC content of the soil [46]. The negative relationship between SOC accumulation and the FQ of LT is a result of the low diversity of thicker-leaved species at our site. This led to a paucity of litter derived from thick leaves, which tend to decompose slowly [31]. As a result, the divergence of LT had a negative impact on SOC storage at the study site. By contrast, the dominance of SLA had a significant, positive association with SOC storage. The vegetation structure along the forest edge is similar to that of secondary forests, with an abundance of species with high SLA [47,48]. SLA is associated with acquisitive trait syndromes in leaves, and promotes carbon loss through leaf decomposition, thus increasing carbon storage in the soil [8,49]. As a result, the dominance of SLA was positively associated with soil carbon storage. This finding confirms the results of other studies that have suggested that SLA may promote increased SOC storage [50,51].

TEC storage was best predicted by increases in the FQ of WD and the CWM of SLA. This indicates that communities with high divergence in WD, combined with an abundance of

species with high SLA, promote increased overall carbon storage. Increases in total carbon storage had a significant, positive association with divergence in WD, which tends to affect AGBC, and were also associated with higher dominance of SLA, which tends to affect SOC. This finding confirms the results of other studies, which report that increasing divergence in WD promotes carbon accumulation at the community level [5,42,52], whereas the dominance of species with high SLA enhances rapid carbon release into the soil [50,51,53]. Our results confirm that carbon accumulation in chronically disturbed DDF is a result of both conservative traits (i.e., dense wood) and acquisitive traits (i.e., high SLA).

## 5. Conclusions

We examined the relationship between functional diversity and carbon storage at the edge of a DDF in northern Thailand. Our results indicate that all carbon storage components were governed by the combined contribution of the CWM and FQ of plant functional traits, i.e., SLA, LT, and WD. They also suggest that carbon storage in chronically disturbed DDF fragments is driven by functional diversity (i.e., dominance and divergence). These findings indicate that understanding the relationship between carbon storage and plant functional traits in DDF edge ecosystems requires the integration of functional dominance (CWM) and divergence (FQ) for all carbon storage components. This study advances our understanding of the mechanisms affecting carbon storage in tropical deciduous forests and can help predict the response of these forests to climate change.

**Supplementary Materials:** The following are available online at https://www.mdpi.com/article/10.3390/su132011416/su132011416/s1, Figure S1: The residuals over predicted values of pairwise relationships between components of carbon storage.

**Author Contributions:** All authors significantly contributed to this paper. Conceptualization, L.A.; methodology, L.A., R.T. and T.K.; software, T.K.; validation, L.A.; formal analysis, R.T; writing—original draft preparation, L.A.; writing—review and editing, L.A; supervision, L.A. All authors have read and agreed to the published version of the manuscript.

**Funding:** This study was supported by funds provided by the Biodiversity-based Economic Development Office (contract number BEDO-NRCT. 21/2017).

**Institutional Review Board Statement:** Not applicable.

**Informed Consent Statement:** Not applicable.

**Data Availability Statement:** Restrictions apply to the availability of these data. Data are available with the permission of the Biodiversity-based Economic Development Office.

**Acknowledgments:** This research was made possible by the assistance of students from the Department of Agroforestry, Phrae Campus, Maejo University. We thank the academic and research staff of Phrae Campus for allowing us to conduct this study.

**Conflicts of Interest:** The authors declare no conflict of interest.

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
