# Peer review of "Plant Functional Diversity Is Linked to Carbon Storage in Deciduous Dipterocarp Forest Edges in Northern Thailand"

_sustainability, doi:10.3390/su132011416_

Round 1

Reviewer 1 Report

Journal: Sustainability (ISSN 2071-1050)

Manuscript ID: sustainability-1375233

Title: Plant Functional Diversity Enhances Carbon Storage in Deciduous Dipterocarp Forest Edges in Northern Thailand

Overall  Comments and Suggestions for Authors

Dear authors,

Regarding carbon storage and plant functional diversity in Thailand, this manuscript could be one of the interesting references, especially for forest and plant science with carbon issues. Generally, I consider this manuscript is described in detail especially in Materials and Method section. The major points that should’ve improved is the results about modeling interpretation. I offer several suggestions authors can consider. I hope that this manuscript can be improved based on peer-review’s comments. My specific comments are as follows.

Point 1.

I agreed most of the approach authors applied for modeling. However, have authors checked about multicollinearity such as Variance Inflation Factor? When a multiple linear regression model is applied, this kind of examination is important to make it stable and unbiased.

Point 2.

As well as Figure 1, I highly suggest authors present the fitted scatterplots such as residuals over predicted values and/or predictors. This is an important step to check the model’s bias. At least, it must be attached as supplementary files in order for the model to be persuasive.

Point 3.

Although I appreciate the data correction and authors’ efforts on it, the model performance such as R-squared is not so high enough. Also, the number of samples are not so sufficient. Therefore, I rather recommend authors restate the argument, which can imply strong impact, e.g. “Enhance” in title. It will express the authors’ opinion in a mild, objective way.

Point 4.

I would like to recommend authors to add a Table about summary statistics of measured trees and plots such as DBH, tree height, weights and etc. as well as Table 1. In addition to AGBC, SOC, and TEC, this general information can help to better understand the stand condition.

Point 5.

In Tables 1, 2, 3 and Figure 1, authors must describe all the abbreviations as far as I know according to the journal guideline. Thus, considering this instruction, writing the variables without abbreviation or acronym can be a better option.

Kind regards,

Reviewer

Author Response

Dear Reviewer 1

Thank you very much for your review on our paper. The comments and suggestions are very useful to improve the manuscript. My responses comments are as follows below.

Comments and Suggestions

Dear authors, regarding carbon storage and plant functional diversity in Thailand, this manuscript could be one of the interesting references, especially for forest and plant science with carbon issues. Generally, I consider this manuscript is described in detail especially in Materials and Method section. The major points that should’ve improved is the results about modeling interpretation. I offer several suggestions authors can consider. I hope that this manuscript can be improved based on peer-review’s comments. My specific comments are as follows.

Comments: Point 1.

I agreed most of the approach authors applied for modeling. However, have authors checked about multicollinearity such as Variance Inflation Factor? When a multiple linear regression model is applied, this kind of examination is important to make it stable and unbiased.

Response: I selected variables for inclusion in stepwise multiple regression that had Pearson correlation coefficients of < 0.7, and tested for the variance inflation factor using the step AIC function of the MASS package in R. Line 262-264 in Track Change file.

Comments: Point 2.

As well as Figure 1, I highly suggest authors present the fitted scatterplots such as residuals over predicted values and/or predictors. This is an important step to check the model’s bias. At least, it must be attached as supplementary files in order for the model to be persuasive.

Response: I agree, fitted the residuals over predicted values plot and added the supplementary 1

Comments: Point 3.

Although I appreciate the data correction and authors’ efforts on it, the model performance such as R-squared is not so high enough. Also, the number of samples are not so sufficient. Therefore, I rather recommend authors restate the argument, which can imply strong impact, e.g. “Enhance” in title. It will express the authors’ opinion in a mild, objective way.

Response: I agree and opinion in a mild, objective way by change enhance to linked in title.

Comments: Point 4.

I would like to recommend authors to add a Table about summary statistics of measured trees and plots such as DBH, tree height, weights and etc. as well as Table 1. In addition to AGBC, SOC, and TEC, this general information can help to better understand the stand condition.

Response: I agree and added trees dimension as DBH, tree height, and biomass in Table 1.

Comments: Point 5.

In Tables 1, 2, 3 and Figure 1, authors must describe all the abbreviations as far as I know according to the journal guideline. Thus, considering this instruction, writing the variables without abbreviation or acronym can be a better option.

Response: I have described all of abbreviations in Tables 1, 2, 3 and Figure 1.

Reviewer 2 Report

Title: Plant Functional Diversity Enhances Carbon Storage in Deciduous Dipterocarp Forest Edges in Northern Thailand
Abstract: 
I find this section fairly okay as it captures the entire research done. 
Introduction:
I also find this section also fairly good but will suggest the content is lengthened a bit to add more background to it (example on forest edge or ecotone relations to carbon storage).
Materials and Methods: 
2.1: Study area: I suggest adding some photos and site maps
2.2: I also suggest adding some visual interpretations of the belt plots
Results: 
Line 194 – 202: The abbreviations are quite a lot and it makes comprehension quite difficult.
Discussion: The study is fairly well discussed but requires some revision. The discussion was mainly focused on the current study with very little clear comparisons with other similar studies. I suggest adding some texts on forest edge or ecotones. 
Conclusion: The manuscript is fairly well concluded.
Although, the study is interesting and will appeal to readers, I suggest the comments raised (and from all reviewers) are duly addressed to make it more comprehensible and concise to bring the quality to a publishable level.

Author Response

Dear Reviewer 2

Thank you very much for your review on our paper. The comments and suggestions are very useful to improve the manuscript. My responses comments are as follows below.

Comments and Suggestions

Title: Plant Functional Diversity Enhances Carbon Storage in Deciduous Dipterocarp Forest Edges in Northern Thailand

Comments:

Abstract: 
I find this section fairly okay as it captures the entire research done.

Response: Thank you very much for your opinion.

Comments:

Introduction:
I also find this section also fairly good but will suggest the content is lengthened a bit to add more background to it (example on forest edge or ecotone relations to carbon storage).

Response: I agree and added the sentence to explain forest edge or ecotone relations to carbon storage (citation number 23 and 24) at line 66-69 in track change file.

Comments:

Materials and Methods: 
2.1: Study area: I suggest adding some photos and site maps

Response: I agree and added site map in Figure 1.

2.2: I also suggest adding some visual interpretations of the belt plots

Response: I agree and added visual interpretations of the belt plots in Figure 2.

Comments:

Results: 
Line 194 – 202: The abbreviations are quite a lot and it makes comprehension quite difficult.

Response: I agree and adding the full words to lines 273-284 in the track change file.

Comments:
Discussion: The study is fairly well discussed but requires some revision. The discussion was mainly focused on the current study with very little clear comparisons with other similar studies. I suggest adding some texts on forest edge or ecotones. 

Response: I agree and added the texts on forest edge citation number 23, 24, 37, 38, and 39 at line 376-381 in track change file.

Comments:
Conclusion: The manuscript is fairly well concluded.

Response: Thank you very much for your opinion.

Comments:
Although, the study is interesting and will appeal to readers, I suggest the comments raised (and from all reviewers) are duly addressed to make it more comprehensible and concise to bring the quality to a publishable level.

Response: Thank you very much for your opinion.

Round 2

Reviewer 1 Report

Journal: Sustainability (ISSN 2071-1050)

Manuscript ID: sustainability-1375233

Title: Plant Functional Diversity Enhances Carbon Storage in Deciduous Dipterocarp Forest Edges in Northern Thailand

In the second rounding of peer-review

Overall  Comments and Suggestions for Authors

Dear authors,

I checked this resubmitted manuscript and the authors’ response. I am glad that authors warmly understood my previous questions. Moreover, I appreciate authors’ efforts to revise the manuscript. I consider Authors have been carefully dealt with all issues I mentioned earlier and updated well enough with additional contents. Also, the supplementary files made me think the models are not biased that much.

As you can see in the Supplementary file, a few of model types presented some bias, but readers can evaluate the model’s accuracy by themselves. Also, all contents were clearer than before.

The resubmitted version is now acceptable and there is no further questions or any comments. Thank you for authors’ contribution on this topic.

Kind regards,

Reviewer